

# Time to get our four priorities right: an 8-year prospective investigation of 1326 player-seasons to identify the frequency, nature, and burden of time-loss injuries in elite Gaelic football

Mark Roe[1], John C. Murphy[2], Conor Gissane[3] and Catherine Blake[1]

[1] School of Public Health, Physiotherapy, and Sports Science, University College Dublin, Dublin, Ireland
[2] Medfit Proactive Health, Dublin, Ireland
[3] School of Sport Health and Applied Science, St Mary's University, Twickenham, London, UK

Corresponding author
Mark Roe, mark.roe@ucd.ie

## ABSTRACT

Managing injury risk requires an understanding of how and when athletes sustain certain injuries. Such information guides organisations in establishing evidence-based priorities and expectations for managing injury risk. In order to minimise the impact of sports injuries, attention should be directed towards injuries that occur frequently, induce substantial time-loss, and elevate future risk. Thus, the current study aimed to investigate the rate at which elite Gaelic football players sustain different time-loss injuries during match-play and training activities. Datasets ($n = 38$) from elite Gaelic football teams ($n = 17$) were received by the National Gaelic Athletic Association Injury Surveillance Database from 2008 to 2016. A total of 1,614 time-loss injuries were analysed. Each season teams sustained 24.0 (interquartile ranges) (IQR 16.0–32.0) and 15.0 (IQR 10.0–19.0) match-play and training injuries, respectively. When exposure was standardised to 1,000 h, greater rates of injury (12.9, 95% CI [11.7–14.3]) and time-loss days (13.4, 95% CI [12.3–14.9]) were sustained in match-play than in training. Acute injury rates were 3.1-times (95% CI [2.7–3.4]) greater than chronic/overuse injuries. Similarly, non-contact injury rates were 2.8-times (95% CI [2.5–3.2]) greater than contact injuries. A total of 71% of injuries in elite Gaelic football affected five lower limb sites. Four lower limb-related clinical entities accounted for 40% of all time-loss injuries (hamstring, 23%; ankle sprain, 7%; adductor-related, 6%; quadriceps strain, 5%). Thus, most risk management and rehabilitation strategies need to be centred around five lower limb sites—and just four clinical entities. Beyond these, it may be highly unlikely that reductions in injury susceptibility can be attributed to specific team interventions. Thus, compliance with national databases is necessary to monitor injury-related metrics and future endeavours to minimise injury risk.

## INTRODUCTION

Gaelic football is a national sport of Ireland and has been governed by the Gaelic Athletic Association (GAA) since 1884. Match-play is characterised by intermittent bouts of multidirectional running as elite players reportedly cover 9,200 m, with 18% at a high-speed pace ($>17 \text{ km} \cdot \text{h}^{-1}$) (*Malone et al., 2016*). This equates to a relative distance of $132 \text{ m} \cdot \text{min}^{-1}$, however during periods of match-play workloads can range between 190 and $230 \text{ m} \cdot \text{min}^{-1}$ (*Malone et al., 2017b*).

Managing injury risk is essential for maximising player availability and team performance (*Hägglund et al., 2013*). The initial stage of this process involves establishing an injury profile for the given sport whilst accounting for the dynamic interactions between players and the activities they undertake (*Auchincloss & Diez Roux, 2008*; *Krieger, 2003*). Thus, managing injury risk requires an understanding of how (i.e. inciting mechanism) and when (i.e. inciting activity) athletes sustain certain injuries (e.g. hamstring strain) (*Roe et al., 2017b*). In order to minimise the impact of injuries in sports, attention can then be directed towards injuries that occur frequently, induce substantial time-loss, and elevate future risk. Such information guides sports organisations in establishing evidence-based priorities by being awareness of 'what problems need to be focused on' when creating future strategies (*Bahr, Clarsen & Ekstrand, 2017*).

For instance, a lot of research and media attention is devoted to anterior cruciate ligament (ACL) injuries, partially due to risk of developing osteoarthritis early in life (*Khan et al., 2018*). However, ACL injuries only account for 2% of all injuries in elite Gaelic football (*Roe et al., 2017a*). Although these infrequent injuries result in an average of 300 days from sport (*Roe et al., 2017a*), 83% of elite athletes return to performance levels comparable to their uninjured peers (*Lai et al., 2018*). Considering that 98% of injuries will not involve the ACL, teams may not experience an ACL injury for two seasons, making it impossible to evaluate the efficacy of specific risk management strategies at a single-team level. Therefore, a need exists for injury surveillance data to support evaluations of team programs via comparisons to epidemiological data on specific clinical entities. In this way, stakeholders may move closer to consensus on what is an acceptable level of risk given awareness of evidence that is relevant, valid, and reliable (*Quarrie et al., 2017*).

Additionally, injury risk management can only be guided with detailed reporting on specific clinical entities. This approach has been eluded to in relation to groin pain in athletes, however, it has not yet been expanded to an injury surveillance dataset encompassing all musculoskeletal injuries (*Weir et al., 2015*). Furthermore, the injury profile of specific activities has yet to be compared in Gaelic football. Thus, the current study aims to establish the frequency, nature, and burden of time-loss injuries sustained in elite male Gaelic football.

## METHODS

Three fundamental variables in epidemiological investigations are the interactions between person, place, and time (*Krieger, 2003*). In a sporting context these can be adapted

to provide an understanding of how (e.g. mechanism) and when (e.g. training, match-play) athletes of a given age sustain specific injuries (e.g. ankle sprain). In the current study we apply these criteria to describe the pattern of injured body regions among elite Gaelic football players. Players are stratified into one of four groups according to age (18–20 years, 21–24 years, 25–29 years, >30 years). The activity during which the injury was sustained (i.e. training or match-play) indicated place. Timing of injury was classified as per seasonal cycle, that is, preseason (weeks 1–7), competitive cycle one (i.e. National League) (weeks 8–16), mid-season (weeks 17–22), or competitive cycle two (i.e. Provincial–National Championship) (weeks 23–34).

Between 2008 and 2016, 38 datasets were received from elite male Gaelic football teams ($n = 17$) enrolled in the National GAA Injury Surveillance Database. This equates to 1,326 player-seasons. The involvement of each team ranged from one to seven seasons. Following consent, player anonymity was maintained and data protection assured in accordance with ethical approval received from the Human Subjects Research Ethics Committee (LS-E-11-91) at University College Dublin. The team medical doctor or Chartered Physiotherapist was responsible for injury diagnosis (Figs. 1 and 2). Team medical staff were asked to confirm whether all injury and exposure data had been provided before reports were generated. Non-compliant teams were then excluded from analysis.

## Definitions

Data were categorised as previously described (*Murphy et al., 2012*). Injury was defined as 'any injury that prevents a player from taking a full part in all training and match play activities typically planned for that day, where the injury has been there for a period greater than 24 h from midnight at the end of the day that the injury was sustained' (*Brooks et al., 2005*). A clinical diagnosis was also selected from a list or entered in free text form and later recoded to defined clinical entities or 'other' if appropriate. Date of partial fitness was defined as 'the date the player is able to participate in training, but is not available for match selection.' Date of full fitness was defined as 'when the player has been able to take a full part in training and is available for match selection.'

## Data analysis

Data were analysed as previously described (*Roe et al., 2016b*) using a statistical analysis software (IBM SPSS Statistics 24.0). Continuous variables are reported as mean with 95% confidence intervals (95% CI). Team rates are reported as median with interquartile ranges (IQR). Injury incidences are reported per 1,000 exposure hours. Injury burden (i.e. time-loss days per 1,000 exposure hours) was calculated by multiplying mean time-loss by the injury incidence. Incidence rate ratios (IRR) were calculated to compare injury risk across age groupings, injury types, and match-play and training activities. IRRs were calculated by dividing a specific incidence metric to that representing the injuries in all other sub-groups.

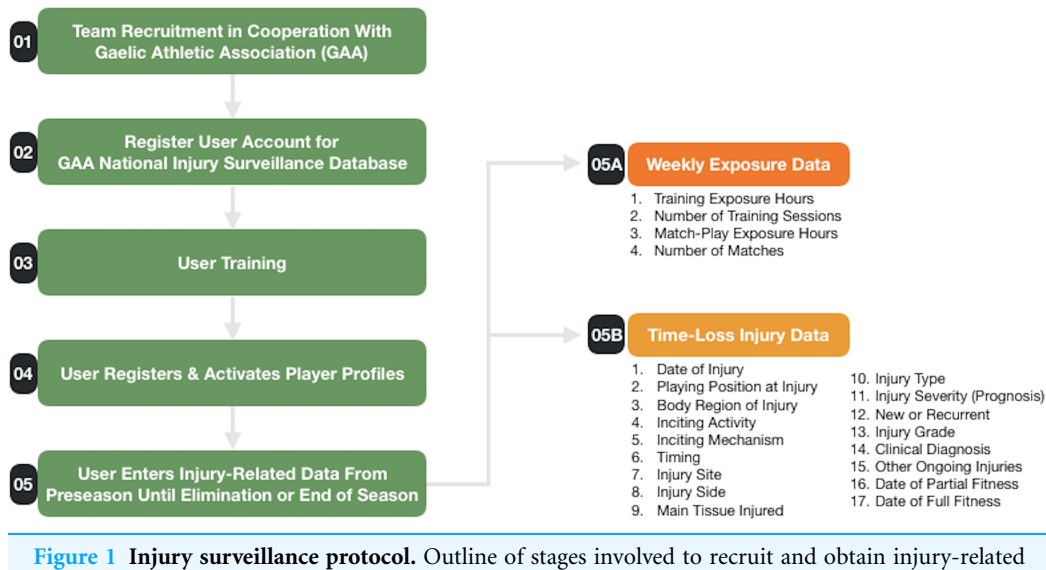

**Figure 1 Injury surveillance protocol.** Outline of stages involved to recruit and obtain injury-related data from team physiotherapists and medical doctors for the GAA National Injury Surveillance Database in 2008–2016.

## RESULTS

A total of 177,854 exposure hours (17,988 match-play; 159,866 training) were reported. Time-loss injuries ($n = 1,606$) were reported for match-play ($n = 896$) and training activities ($n = 616$). An additional 94 time-loss injuries (5.9%, 95% CI [4.7–7.1]) were associated with an insidious onset as opposed to a specific activity.

### Team rates

The median number of injuries sustained per team each season was 42.0 (IQR 31.0–53.0). Each season teams sustained 24.0 (IQR 16.0–32.0) and 15.0 (IQR 10.0–19.0) match-play and training injuries, respectively. In total, 33.0 (IQR 22.0–45.0) injuries were sustained during competitive cycles (Table 1).

### Injury site

Match-play was associated with the onset of 54.4% (95% CI [51.7–57.2]) lower limb, 75.6% (95% CI [69.0–82.1]) upper limb, 41.9% (95% CI [32.4–51.4]) trunk, and 62.2% (95% CI [48.9–75.6]) head/neck injuries. Training was associated with the onset of 40.3% (95% CI [37.5–48.2]) lower limb, 23.2% (95% CI [16.7–29.8]) upper limb, 43.8% (95% CI [34.3–53.3]) trunk, and 33.3% (95% CI [20.0–48.9]) head/neck injuries.

   The five most common injuries were lower limb related and accounted for 70.9% (95% CI [62.4–78.9]) of all time-loss injuries. However, the frequency of these common injuries differed between match-play and training (Table 2). Analysis of clinical entities revealed that four specific injuries accounted for 40.9% (95% CI [35.6–46.1]) of all injuries. These related to hamstring strains (23.0%), ankle sprains (6.8%), adductor-related groin pain (5.9%), and quadriceps strains (5.2%). Quartile ranges identified that aside from these

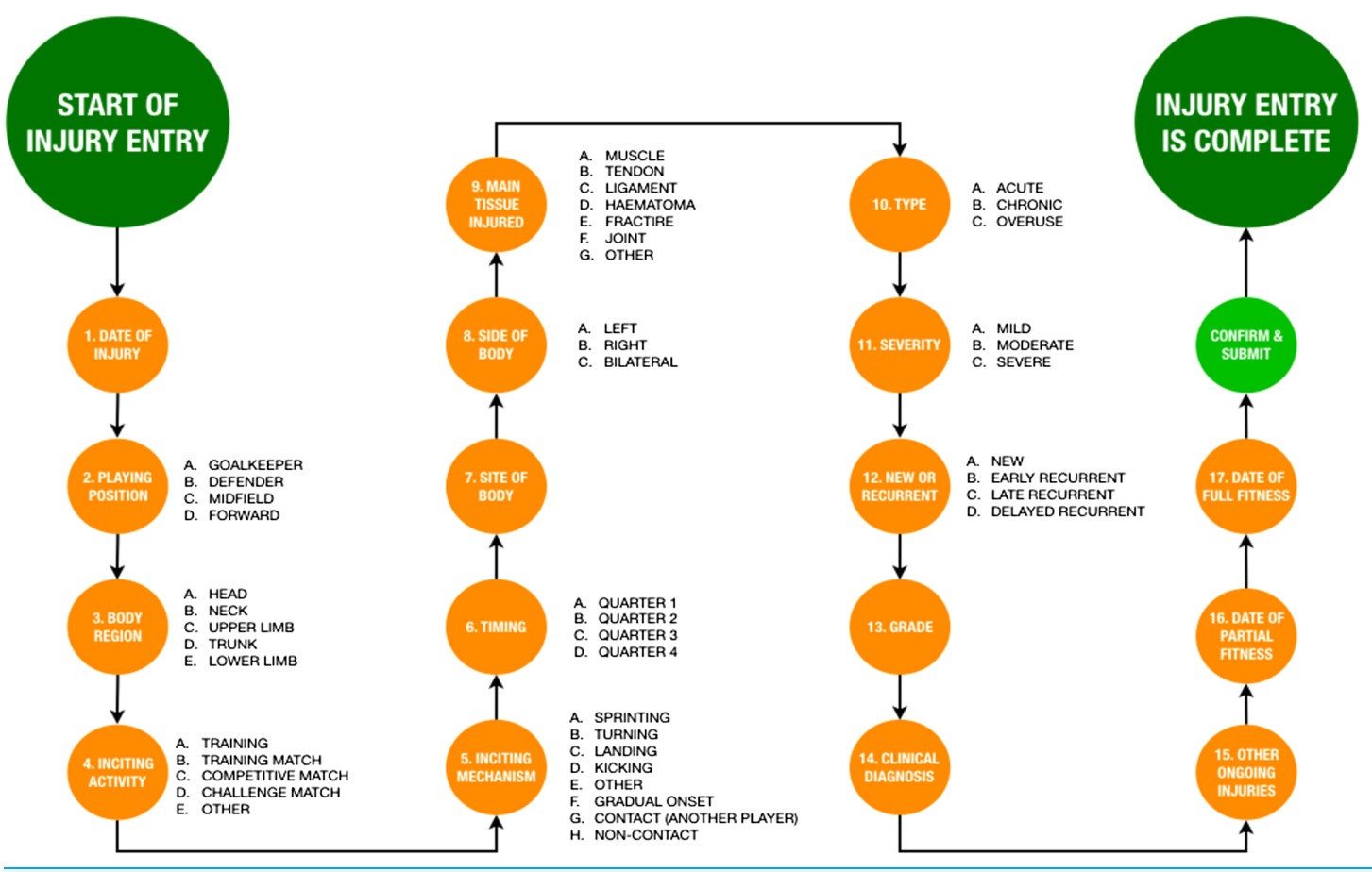

**Figure 2 Data entry pathway for registering a time-loss injury.**

four clinical entities, each season more than one-in-four teams will not sustain injuries identified as being the most common (Table 3). These four clinical entities also accounted for 38.9% (95% CI [29.3–52.7]) of all player unavailability (Table 4).

## Match-play injuries

Match-play injuries accounted for 55.8% (53.5–58.2) of all time-loss injuries. The lower limb region was the most common site of match-play injury (77.8%, 95% CI [75.1–80.3]) followed by the upper limb (14.2%, 95% CI [12.1–16.5]), trunk (4.9%, 95% CI [3.5–6.3]), and head/neck regions (3.1%, 95% CI [2.0–4.2]).

Most match-play injuries were classified as new (78.0%, 95% CI [73.3–82.7]) as opposed to recurrent (22.0%, 95% CI [17.3–26.7]), were associated with an acute onset (81.7%, 95% CI [79.2–84.2]) rather than chronic or overuse (18.3%, 95% CI [15.8–20.8]), and were incited by non-contact mechanisms (73.2%, 59.8–65.3) as opposed to contact between players (36.8%, 95% CI [33.7–40.2]).

Match-play was associated with the onset of 55.8% (95% CI [53.4–58.2]) of all new injuries and 59.3% (95% CI [50.4–67.4]) of all recurrent injuries. Furthermore, the proportions of all early (<8 weeks), late (2–12 months), and delayed (>12 months)

**Table 1 Injury rates per team.**

|  | Median | Interquartile range |
|---|---|---|
| **Team rates** |  |  |
| Total injuries | 42.0 | 31.0–53.0 |
| Match-play injuries | 24.0 | 16.0–32.0 |
| Training injuries | 15.0 | 10.0–19.0 |
| **Region** |  |  |
| Lower limb | 33.0 | 24.0–45.0 |
| Upper limb | 4.0 | 3.0–7.0 |
| Trunk | 2.0 | 1.0–5.0 |
| Head/neck | 1.0 | 1.0–2.0 |
| **Injury type** |  |  |
| Contact | 11.0 | 7.0–16.0 |
| Non-contact | 30.0 | 21.0–39.0 |
| Acute | 32.0 | 26.0–36.0 |
| Chronic/overuse | 9.0 | 5.0–13.0 |
| New | 31.0 | 23.0–43.0 |
| Recurrent | 9.0 | 7.0–11.0 |
| **Severity** |  |  |
| Mild | 11.0 | 6.0–17.0 |
| Moderate | 19.0 | 11.0–26.0 |
| Severe | 10.0 | 6.0–12.0 |
| **Seasonal cycle** |  |  |
| Preseason | 4.0 | 2.0–7.0 |
| Competitive cycle 1 | 20.0 | 14.0–29.0 |
| Midseason | 7.0 | 4.0–9.0 |
| Competitive cycle 2 | 13.0 | 8.0–16.0 |

**Note:**
Presented as median (interquartile range) per season.

recurrent injuries occurring in match-play were 59.3% (95% CI [50.4–67.4]), 44.6% (95% CI [36.7–52.5]), and 44.6% (95% CI [32.5–55.4]), respectively.

Analysis of time-loss data revealed that 25.1% (95% CI [21.7–28.5]), 51.1% (95% CI [47.5–55.4]), and 23.8% (95% CI [20.0–27.0]) of match injuries resulted in mild, moderate, and severe time-loss, respectively.

## Training injuries

Training injuries most commonly occurred in the lower limb region (83.6%, 95% CI [80.5–86.5]) followed by the trunk (7.5%, 95% CI [5.4–9.4]), upper limb (6.3%, 95% CI [4.2–8.1]), and head/neck regions (2.4%, 95% CI [1.3–3.9]).

Training was associated with the onset of 38.7% (36.3–41.2) of all new injuries and 35.6% (95% CI [28.1–43.7]) of all recurrent injuries. Furthermore, the proportions of early (<8 weeks), late (2–12 months), and delayed (>12 months) recurrent injuries that occurred during training were 35.6% (95% CI [28.1–43.7]), 46.8% (95% CI [38.1–54.7]), and 47.0% (95% CI [36.1–59.0]), respectively.

**Table 2 Five most commonly injured sites.**

| | All injuries | Match-play | Training |
|---|---|---|---|
| 1 | Hamstring, 23.9% (21.9–26.0) | Hamstring, 23.1% (20.2–26.0) | Hamstring, 27.6% (24.2–31.2) |
| 2 | Groin, 14.9% (13.0–16.7) | Knee, 12.7% (10.5–15.0) | Groin, 17.5% (14.8–20.6) |
| 3 | Ankle, 11.7% (10.1–13.3) | Ankle, 12.2% (9.9–14.2) | Ankle, 10.7% (8.4–13.3) |
| 4 | Knee, 11.1% (9.5–12.6) | Groin, 10.8% (8.9–12.9) | Quadriceps, 10.1% (7.8–12.5) |
| 5A | Quadriceps, 9.3% (7.9–10.3) | Shoulder, 9.7% (7.9–11.7) | Knee, 8.6% (6.3–10.7) |
| 5B | – | Quadriceps, 9.7% (7.7–11.6) | – |
| Combined | 70.9% (62.4–78.9) | 78.2% (65.1–91.4) | 74.5% (61.5–88.3) |

**Note:**
Presented with corresponding 95% confidence intervals.

Time-loss data revealed that 30.2% (95% CI [25.8–34.9]), 50.4% (95% CI [45.7–55.8]), and 19.4% (95% CI [15.5–23.3]) of training injuries resulted in mild, moderate, and severe time-loss, respectively.

### Injury incidence across age-groups

Overall injury incidence was 9.2 (95% CI [8.8–9.6]) per 1,000 exposure hours. The incidence of acute injuries was 3.1-times (95% CI [2.7–3.4]) greater than chronic/overuse injuries. Similarly, non-contact injuries occurred 2.8-times (95% CI [2.5–3.2]) more frequently than injuries incited via contact between players. Injury incidence increased across age-groups with IRR greatest for players aged 30+ years (1.51, 95% CI [1.32–1.74]) when compared to all other players (Table S1).

### Injury incidence between match-play and training

Match-play incidence (49.8, 95% CI [46.5–53.0]) was 12.9-times (95% CI [11.7–14.3]) higher than training incidence (3.9, 95% CI [3.6–4.3]). The incidence and IRR of non-contact, contact, acute, and chronic/overuse injuries between match-play and training are outlined in Tables 5 and 6.

The incidence of mild, moderate, and severe injuries was also compared between match-play and training (Table 7) whilst accounting for injury region. The IRR between match-play and training injuries grew as the classification of severity increased (Table 6).

### Time-loss and player unavailability

Match-play injuries accounted for a greater proportion of all time-loss (51.8%, 95% CI [51.2–52.1]) than training injuries (33.8%, 95% CI [3.4–36.6]). Match-play and training injuries resulted in a total of 576 (95% CI [345.6–851.2]) and 342 (95% CI [193.0–516.8]) time-loss days per team each season, respectively. The mean time-loss for match-play and training injuries was 24.0 (95% CI [21.6–26.6]) and 22.8 (95% CI [19.3–27.2]) days, respectively. Although mean time-loss overlaps considerably between these activities, injury burden (days lost per 1,000 h of exposure) was 13.4-times (95% CI [12.3–14.9]) higher in match-play than in training (Table 8). Lower limb injuries accounted for the majority (79.3%, 95% CI [77.1–80.8]) of player unavailability. This trend was observed across all age-groups (Table S2).

**Table 3 Most common clinical entities as per injury diagnosis.**

| | Team rate per season (median, IQR) | Percentage of all injuries | Prevalence | Incidence | Match-play incidence | Training incidence | Likelihood non-contact related | Likelihood occuring in match-play | Likelihood occuring in training |
|---|---|---|---|---|---|---|---|---|---|
| All injuries | 42 (31–53) | – | 69.8% (67.4–72.3) | 9.2 (8.8–9.6) | 50.5 (47.2–53.8) | 3.9 (3.6–4.3) | 72.9% (70.7–75.1) | 55.8% (53.4–58.2) | 38.7% (36.3–41.2) |
| Hamstring: muscle/tendon strain | 9 (7–12) | 23.0% (21.0–24.9) | 19.7% (17.5–21.8) | 2.1 (1.9–2.3) | 11.0 (9.5–12.5) | 1.1 (0.9–1.2) | 98.1% (96.8–99.5) | 52.7% (48.1–57.7) | 45.2% (40.4–49.7) |
| Ankle sprain | 3 (1–4) | 6.8% (5.6–7.9) | 7.4% (6.0–8.8) | 0.6 (0.5–0.7) | 3.8 (2.9–4.7) | 0.2 (0.2–0.3) | 80.2% (72.1–87.4) | 62.2% (53.2–71.2) | 32.4% (23.4–40.5) |
| Groin: adductor-related | 2 (1–4) | 5.9% (4.8–6.9) | 6.0% (4.7–7.2) | 0.5 (0.4–0.7) | 2.5 (1.8–3.3) | 0.3 (0.2–0.4) | 94.8% (89.6–99.0) | 47.9% (37.9–57.9) | 46.9% (36.9–56.9) |
| Quadriceps: muscle strain | 2 (1–4) | 5.2% (4.2–6.4) | 5.8% (4.5–7.1) | 0.5 (0.4–0.6) | 2.0 (1.3–2.6) | 0.3 (0.2–0.4) | 98.8% (96.5–100) | 41.2% (30.6–50.6) | 58.8% (49.4–69.4) |
| Calf: muscle/tendon strain | 1 (0–4) | 4.3% (3.4–5.3) | 4.5% (3.4–5.6) | 0.4 (0.3–0.5) | 2.0 (1.3–2.7) | 0.2 (0.1–0.3) | 94.4% (87.3–98.6) | 50.7% (38.0–62.0) | 47.9% (36.6–60.6) |
| Quadriceps: bruising/ haematoma | 1 (0–3) | 3.9% (3.0–4.8) | 4.1% (3.1–5.2) | 0.4 (0.3–0.4) | 2.8 (2.1–3.6) | 0.1 (0.0–0.1) | 7.9% (1.6–15.9) | 81.0% (71.4–90.5) | 17.5% (7.9–27.0) |
| Groin: hip-related | 1 (0–3) | 2.7% (2.0–3.5) | 2.9% (2.0–3.9) | 0.3 (0.2–0.3) | 0.9 (0.5–1.3) | 0.1 (0.1–0.2) | 68.2% (54.5–81.8) | 35.6% (21.6–49.5) | 46.7% (32.1–61.2) |
| Shoulder AC joint sprain | 1 (0–2) | 2.4% (1.7–3.1) | 2.9% (2.0–3.8) | 0.2 (0.2–0.3) | 1.8 (1.2–2.5) | 0.04 (0.0–0.1) | 15.0% (5.0–27.4) | 82.5% (70.0–92.5) | 15.0% (5.0–27.5) |
| Ankle: general | 1 (0–2) | 2.4% (1.7–3.1) | 2.7% (1.8–3.6) | 0.2 (0.2–0.3) | 1.1 (0.6–1.6) | 0.1 (0.1–0.1) | 76.9% (64.1–89.7) | 51.3% (33.3–66.7) | 41.0% (25.6–56.4) |
| Groin: other | 1 (0–3) | 2.3% (1.6–3.1) | 2.6% (1.7–3.4) | 0.2 (0.2–0.3) | 0.7 (0.3–1.0) | 0.1 (0.1–0.2) | 89.5% (78.9–97.4) | 31.6% (16.8–46.4) | 47.4% (31.5–63.2) |
| Groin: Iliopsoas-related | 1 (0–3) | 2.1% (1.5–2.9) | 2.5% (1.7–3.3) | 0.2 (0.1–0.3) | 0.8 (0.4–1.3) | 0.1 (0.1–0.2) | 88.6% (77.1–97.1) | 42.9% (26.5–59.3) | 48.6% (32.0–65.1) |
| Knee: patellar tendinopathy | 0 (0–1) | 2.1% (1.4–2.8) | 2.3% (1.5–3.1) | 0.2 (0.1–0.3) | 1.3 (0.8–1.8) | 0.03 (0.0–0.1) | 97.1% (91.2–100) | 67.6% (52.9–82.4) | 11.8% (2.9–23.5) |
| Shoulder: general | 0 (0–1) | 1.7% (1.2–2.3) | 2.1% (1.3–2.9) | 0.2 (0.1–0.2) | 1.0 (0.5–1.5) | 0.06 (0.0–0.1) | 25.0% (7.1–42.9) | 64.3% (46.4–82.1) | 35.7 (17.9–53.6) |
| Knee: MCL sprain | 1 (0–2) | 1.6% (1.0–2.3) | 1.7% (1.0–2.4) | 0.2 (0.1–0.2) | 1.2 (0.7–1.7) | 0.03 (0.0–0.1) | 42.3% (23.1–61.5) | 80.8% (65.4–96.2) | 15.4% (3.8–30.8) |
| Knee: general | 0 (0–1) | 1.6% (1.0–2.2) | 2.0% (1.2–2.7) | 0.2 (0.1–0.2) | 0.6 (0.3–1.0) | 0.1 (0.0–0.1) | 61.5% (42.4–80.8) | 42.3% (23.1–61.5) | 53.8% (34.6–73.1) |
| Back: disc pathology | 0 (0–1) | 1.5% (1.0–2.1) | 1.7% (1.0–2.3) | 0.2 (0.1–0.2) | 0.4 (0.1–0.8) | 0.1 (0.0–0.1) | 96.0% (88.0–100) | 32.0% (16.0–52.0) | 44.0% (24.0–64.0) |
| Knee: bruising/ haematoma | 0 (0–1) | 1.5% (0.9–2.2) | 2.5% (1.7–3.3) | 0.2 (0.1–0.3) | 1.3 (0.8–1.8) | 0.06 (0.0–0.1) | 37.1% (22.9–54.3) | 65.7% (48.6–82.8) | 25.7% (11.4–40.0) |

**Table 4 Consequences of the most common clinical entities.**

| | Mean time-loss | Injury burden | Percentage of unavailability | Likelihood of recurrence |
|---|---|---|---|---|
| All injuries | 25.9 (23.5–28.4) | 238.3 (206.8–272.6) | – | 71.8% (63.5–80.2)* |
| Hamstring: muscle/tendon strain | 25.2 (20.5–31.0) | 52.9 (39.0–71.3) | 22.2% (18.8–26.2) | 44.1% (38.0–50.1) |
| Ankle: sprain | 24.5 (18.1–32.9) | 14.7 (9.1–23.0) | 6.2% (4.4–8.4) | 13.3% (6.5–20.0) |
| Groin: adductor-related | 25.3 (15.9–37.1) | 12.7 (6.4–26.0) | 5.3% (3.1–9.5) | 21.5% (12.5–30.6) |
| Quadriceps: muscle strain | 24.7 (15.5–38.9) | 12.4 (6.2–23.3) | 5.2% (3.0–8.6) | 10.4% (3.6–17.2) |
| Calf: muscle/tendon strain | 29.2 (19.2–42.3) | 11.7 (5.8–21.2) | 4.9% (2.8–7.8) | 18.3% (8.5–28.1) |
| Quadriceps: bruising/haematoma | 9.7 (7.8–11.9) | 3.9 (2.3–4.8) | 1.6% (1.1–1.7) | 14.5% (5.2–23.9) |
| Groin: hip-related | 32.1 (19.0–47.3) | 9.6 (3.8–14.2) | 4.0% (1.8–5.2) | 12.8% (2.3–23.3) |
| Shoulder AC joint sprain | 34.2 (20.0–54.0) | 6.8 (4.0–16.2) | 2.9% (1.9–5.9) | 5.3% (1.8–12.4) |
| Ankle: general | 26.8 (17.7–38.3) | 5.4 (3.5–11.5) | 2.2% (1.7–4.2) | 8.3% (0.7–17.4) |
| Groin: other | 18.1 (9.7–29.1) | 3.6 (1.9–8.7) | 1.5% (0.9–3.2) | 11.8% (0.9–22.6) |
| Groin: Iliopsoas-related | 13.3 (10.1–17.1) | 2.7 (1.0–5.1) | 1.1% (0.5–1.9) | 6.1% (2.1–14.2) |
| Knee: patellar tendinopathy | 41.7 (22.7–64.3) | 8.3 (2.3–19.3) | 3.5% (1.1–7.1) | 20.0% (5.7–34.3) |
| Shoulder: general | 15.8 (10.9–20.8) | 3.2 (1.1–4.2) | 1.3% (0.5–1.5) | None registered |
| Knee: MCL sprain | 32.2 (22.0–43.4) | 6.4 (2.2–8.7) | 2.7% (1.1–3.2) | 13.0% (0.7–26.8) |
| Knee: general | 19.5 (12.1–27.3) | 3.9 (1.2–5.5) | 1.6% (0.6–2.0) | None registered |
| Back: disc pathology | 41.9 (12.9–85.1) | 8.4 (1.3–17.0) | 3.5% (0.6–6.2) | 13.6% (0.7–28.0) |
| Knee: bruising/haematoma | 18.8 (10.7–31.3) | 3.8 (1.1–6.3) | 1.6% (0.5–3.4) | 6.1% (2.1–14.2) |

Note:
* Likelihood of recurrence statistic in row 'All Injuries' refers to proportion of players sustaining a subsequent injury following a return to sport.

**Table 5 Frequency and nature of match-play and training injuries per 1,000 h.**

| | Incidence | Non-contact | Contact | NC: C IRR | Acute | Chronic/overuse | A: C/O IRR |
|---|---|---|---|---|---|---|---|
| **Match-play injuries** | | | | | | | |
| All regions | 49.8 (46.5–53.0) | 29.1 (26.6–31.6) | 21.0 (18.6–22.8) | 1.41 (1.23–1.60) | 41.9 (38.9–38.9) | 7.8 (6.5–9.1) | 5.35 (4.48–6.39) |
| Lower limb | 38.7 (35.8–41.6) | 25.5 (23.2–27.9) | 13.2 (11.5–14.9) | 1.94 (1.66–2.26) | 31.7 (29.1–34.3) | 7.0 (5.8–8.2) | 4.52 (3.74–5.48) |
| Upper limb | 7.1 (5.8–8.3) | 2.4 (1.7–3.1) | 4.7 (3.7–5.7) | 0.51 (0.3–0.74) | 6.6 (5.4–7.7) | 0.5 (0.2–0.8) | 13.11 (6.66–25.81) |
| Trunk | 2.5 (1.7–3.2) | 1.2 (0.7–1.7) | 1.3 (0.8–1.8) | 0.91 (0.51–1.65) | 2.1 (1.4–2.8) | 0.3 (0.1–0.6) | 6.33 (2.68–14.98) |
| Head/neck | 1.6 (1.0–2.1) | – | 1.6 (1.0–2.1) | – | 1.6 (1.0–2.1) | – | – |
| **Training injuries** | | | | | | | |
| All regions | 3.9 (3.5–4.2) | 2.7 (2.4–3.0) | 1.2 (1.0–1.3) | 2.34 (1.97–2.78) | 2.9 (2.7–3.2) | 0.9 (0.8–1.1) | 3.24 (2.69–3.90) |
| Lower limb | 3.2 (2.9–3.5) | 2.4 (2.2–2.7) | 0.8 (0.7–0.9) | 3.02 (2.48–3.69) | 2.4 (2.2–2.7) | 0.8 (0.7–1.0) | 2.99 (2.45–3.65) |
| Upper limb | 0.2 (0.2–0.3) | 0.1 (0.0–0.1) | 0.2 (0.1–0.2) | 0.44 (0.23–0.88) | 0.2 (0.2–0.3) | 0.02 (0.00–0.04) | 12.00 (3.70–38.97) |
| Trunk | 0.3 (0.2–0.4) | 0.2 (0.1–0.2) | 0.1 (0.1–0.2) | 1.71 (0.94–3.10) | 0.2 (0.1–0.3) | 0.1 (0.0–0.1) | 2.54 (1.34–4.82) |
| Head/neck | 0.1 (0.1–0.1) | 0.02 (0.00–0.04) | 0.1 (0.0–0.1) | 0.25 (0.07–0.89) | 0.1 (0.1–0.1) | – | – |

Note:
IRR, Incidence rate ratio.

## DISCUSSION

The aim of the current study was to establish the rate at which elite Gaelic football players sustain different time-loss injuries during match-play and training activities. Measures of

**Table 6 Injuries per 1,000 h in match-play compared to training.**

| | IRR | Non-contact | Contact | Acute | Chronic/overuse | Mild | Moderate | Severe |
|---|---|---|---|---|---|---|---|---|
| All regions | 12.93 (11.69–14.32) | 10.78 (9.50–12.24) | 17.97 (15.07–21.43) | 14.26 (12.72–15.98) | 8.64 (6.86–10.89) | 10.94 (9.00–13.29) | 13.26 (11.48–15.31) | 15.98 (12.78–19.99) |
| Lower limb | 12.01 (10.73–13.45) | 10.54 (9.22–12.06) | 16.46 (13.28–20.39) | 13.12 (11.54–14.92) | 8.68 (6.79–11.09) | 10.50 (8.44–13.05) | 11.78 (1.07–13.79) | 14.90 (11.58–19.18) |
| Upper limb | 28.94 (20.22–41.42) | 31.85 (16.80–60.38) | 27.65 (17.93–42.65) | 29.13 (20.06–42.29) | 26.66 (7.22–98.46) | 33.01 (14.33–76.04) | 23.96 (14.85–38.65) | 61.22 (30.43–123.16) |
| Trunk | 8.50 (5.62–12.85) | 6.44 (3.67–11.28) | 12.02 (6.43–22.50) | 10.23 (6.42–16.31) | 4.10 (1.56–10.79) | 5.43 (2.57–11.50) | 19.39 (9.50–39.50) | 8.00 (3.25–19.68) |
| Head/neck | 16.59 (8.86–31.05) | – | 20.74 (10.55–40.77) | 16.59 (8.86–31.05) | – | 12.22 (4.92–30.38) | 41.47 (11.92–144.31) | 11.85 (2.65–52.94) |

Note:
IRR, Incidence rate ratio.

**Table 7 Time-loss per activity.**

| | Mild | Moderate | Severe | Mild incidence | Moderate incidence | Severe incidence |
|---|---|---|---|---|---|---|
| All injuries | 27.0% (24.4–29.5) | 49.8% (47.0–53.0) | 23.2% (20.6–25.7) | 2.5 (2.2–2.7) | 4.5 (4.2–4.8) | 2.1 (1.9–2.3) |
| Lower limb | 26.6% (23.4–29.5) | 50.8% (47.4–54.2) | 22.6% (19.9–25.6) | 1.9 (1.7–2.1) | 3.7 (3.4–4.0) | 1.6 (1.5–1.8) |
| Upper limb | 19.8% (12.5–28.1) | 51.0% (40.6–60.4) | 29.2% (19.8–38.5) | 0.1 (0.1–0.2) | 0.3 (0.2–0.4) | 0.2 (0.1–0.2) |
| Trunk | 35.8% (25.4–46.3) | 40.3% (28.3–52.2) | 23.9% (13.4–34.3) | 0.2 (0.2–0.3) | 0.2 (0.2–0.3) | 0.1 (0.1–0.2) |
| Head/neck | 44.0% (24.0–64.0) | 40.0% (20.0–60.0) | 16.0% (4.0–32.0) | 0.1 (0.1–0.2) | 0.1 (0.1–0.2) | 0.1 (0.0–0.1) |
| Training injuries | 30.2% (26.0–34.6) | 50.4% (45.4–55.0) | 19.4% (15.7–23.3) | 1.1 (1.0–1.3) | 1.9 (1.7–2.1) | 0.7 (0.6–0.9) |
| Lower limb | 28.9% (24.1–33.7) | 52.4% (46.5–57.5) | 18.7% (15.0–22.9) | 0.9 (0.8–1.1) | 1.7 (1.5–1.9) | 0.6 (0.5–0.7) |
| Upper limb | 17.6% (0.0–35.3) | 58.8% (35.3–82.4) | 23.5% (5.9–41.2) | 0.04 (0.01–0.08) | 0.14 (0.09–0.20) | 0.06 (0.02–0.09) |
| Trunk | 46.4% (28.6–64.3) | 28.6% (10.7–46.4) | 25.0% (10.7–42.9) | 0.11 (0.06–0.16) | 0.07 (0.03–0.11) | 0.06 (0.02–0.10) |
| Head/neck | 55.6% (22.2–88.9) | 22.2% (0.0–55.6%) | 22.2% (0.0–55.6%) | 0.05 (0.02–0.08) | 0.02 (0.00–0.04) | 0.02 (0.00–0.04) |
| Match-injuries | 25.1% (21.7–28.7) | 51.1% (47.0–55.2) | 23.8% (20.3–27.0) | 12.5 (10.8–14.1) | 25.5 (23.1–27.8) | 11.9 (10.3–13.5) |
| Lower limb | 25.3% (21.2–29.2) | 51.5% (47.0–56.3) | 23.2% (19.3–26.8) | 9.8 (8.3–11.2) | 19.9 (17.8–22.0) | 9.0 (7.6–10.3) |
| Upper limb | 20.5% (1.5–29.5) | 48.7% (37.2–60.3) | 30.8% (20.5–41.0) | 1.5 (0.9–2.0) | 3.5 (2.6–4.3) | 2.2 (1.5–2.9) |
| Trunk | 25.0% (10.7–42.9) | 53.6% (35.7–71.4) | 21.4% (7.1–35.7) | 0.6 (0.3–1.0) | 1.3 (0.8–1.9) | 0.5 (0.2–0.8) |
| Head/neck | 37.5% (18.8–62.5) | 50.0% (25.0–75.0) | 12.5% (7.1–35.7) | 0.6 (0.3–1.0) | 0.8 (0.4–1.2) | 0.2 (0.0–0.4) |

Note:
Mild (1–7 days), moderate (8–28 days), severe (29+ days). Incidence reported per 1,000 exposure hours.

**Table 8 Time-loss and injury burden per activity.**

| | Mean time-loss days | | | Percentage of all time-loss | | | |
|---|---|---|---|---|---|---|---|
| | All injuries | Match-play | Training | All injuries | Match-play | Training | Injury burden MP: TR RR |
| All regions | 25.7 (23.4–28.3) | 24.0 (21.6–26.6) | 22.8 (19.3–27.2) | – | 51.8% (51.2–52.1) | 39.5% (36.7–42.8) | 13.44 (12.34–14.87) |
| Lower limb | 25.4 (22.8–28.5) | 23.5 (21.1–26.4) | 22.6 (18.9–27.2) | 79.1% (77.9–80.6) | 76.1% (76.0–77.2) | 83.0% (82.0–83.7) | 12.58 (11.54–13.78) |
| Upper limb | 28.8 (22.5–36.4) | 29.9 (22.3–38.3) | 23.7 (16.8–31.3) | 11.2% (9.6–12.8) | 17.7% (14.6–20.4) | 6.6% (5.5–7.3) | 44.79 (38.49–54.50) |
| Trunk | 28.9 (18.6–42.6) | 22.4 (13.3–35.8) | 27.0 (13.4–45.2) | 7.4% (5.2–9.9) | 4.6% (3.0–6.6) | 8.9% (5.2–12.4) | 6.94 (6.34–8.44) |
| Head/neck | 13.3 (8.8–18.2) | 11.8 (7.8–17.4) | 16.0 (7.5–25.3) | 1.8% (1.3–2.2) | 1.5% (1.1–2.0) | 1.7% (0.9–2.3) | 11.80 (10.4–14.44) |

Note:
MP, Match-play; TR, training; RR, relative ratio.

central tendency reveal that teams sustain 24 match-play and 15 training injuries per season. Injury incidence per 1,000 exposure hours is 12.9-times greater in match-play (49.8/1,000 h) than in training (3.9/1,000 h). Essentially, teams are sustaining the vast majority of their injuries during match-play despite only periodically playing competitive matches. The magnitude of inequity between activity injury rates means that identification of factors influencing the onset of match-play injuries should be prioritised given their greater rate of occurrence.

## Emergence of match-play injury patterns

Despite contact injuries being 18.0-times more frequent in match-play than in training, 73% of match-play injuries were classified as non-contact. Furthermore, most match-play injuries were classified as new (78%) and acute (82%) suggesting an adverse

relationship between player's capacities and imposed match-play demands. Thus, factors such as contact between players and deficits from previous or ongoing conditions are not associated with the onset of most match-play injuries. This prompts the question: why do acute, non-contact injuries occur more frequently in match-play than in training? Although random events impact injury susceptibility, it is unlikely that elite players become 13-times unluckier when participating in than in training.

A greater proportion of recurrent injuries (59%) occurred in match-play than in training (36%), particularly during the immediate eight weeks following return to sport (59% vs 30%). Studies in elite soccer players have observed decrements in lower limb strength following exposure to match-play, particularly among previously injured players (*Small et al., 2010*; *Wollin, Thorborg & Pizzari, 2017*). The odds of sustaining injury are also known to be greater among elite Gaelic football players with previous injuries in comparison to their uninjured peers (*Roe et al., 2016b*). Likewise, elite Gaelic footballers with previous hamstring injuries may have greater eccentric knee flexor strength on average when compared to their uninjured peers, the likelihood of decrements following return to sport was 51% with a 25% chance of between limb asymmetries exceeding 15% (*Roe et al., 2018*). Such findings may guide return to sport protocols and tailoring of risk management strategies among players with recent previous injuries as unique management strategies may be required for this sub-cohort.

## Paradox of performance-focused teams sustaining frequent match-play injuries

Considering that training aims to maximise the chances of the team succeeding in match-play, a high match-play injury rate, largely constituted by non-contact and acute injuries, cannot coexist with interventions to maximise player availability. The high rate of injuries (IQR 14.0–29.0) during the initial competitive cycle of the season suggests components of preseason interventions offer little protection against early in-season exposures to injurious match-play demands. Unavailability may impair the transition of early career players to senior squads by reducing exposures to interventions to develop desired sport specific skills while promote detraining during rehabilitation periods (*Joo, 2016*; *Koundourakis et al., 2014*; *Silva et al., 2016*).

The rate of injuries not associated with contact or chronic/overuse injuries in match-play suggests emergence of an injury pattern distinct to training. This suggests scope for screening studies to detect players especially vulnerable to match-play demands as it is questionable that a truly random series of factors are driving this increased rate. However, identifying athletes at greater risk of sustaining injury, due to modifiable factors, has been challenging to date (*Bahr, 2016*). While it is clear that most injuries in elite Gaelic football are sustained during match-play, investigations of injury risk factors typically occur in preseason and thus, not periods associated with frequent match-play exposures. This leads to a reliance on surrogate and cross-sectional measures of injury risk to assess capacity to tolerate match-play demands for a prolonged period of time. These findings suggest complex dynamics between an athlete's work capacity, tolerance of sport specific stress, and injury.

The workloads imposed on athletes in match-play and training have yet to be compared in elite Gaelic football. However, during training camp sessions players have reportedly covered 5,417 ± 425 m, of which 924.4 ± 225 m was at ≥17 km·hr or high-speed distance (*Malone et al., 2017a*). This is 42% less than the 9,222 ± 1,588 m and 1,596 ± 594 m at ≥17 km·hr, reportedly covered in match-play (*Malone et al., 2016*). Contextual factors such as seasonal cycle, opposition standard, tactical strategies, and match outcome also impact these workloads (*Ryan et al., 2017*). Despite being central to the activity during which most injuries are sustained (i.e. match-play) these factors have yet to be considered by screening tools.

## Interpretating epidemiological data to guide selection of screening tools

Understanding the extent of the injury problem is the first stage of reducing injury risk (*Van Mechelen, Hlobil & Kemper, 1992*). This is the key to designing risk management strategies as it guides researchers and practitioners with an understanding of how, when, and where certain athletes sustain certain injuries (*Roe et al., 2017b*). Results of the current study show that training and match-play have different injury profiles as marked by their distinct common injury sites, inciting mechanisms, types, and severity patterns. Thus, the first stage in designing screening protocols for common injuries needs to consider the exact injury of focus, during which activity and mechanism it mostly occurs, as well as the seasonal cycle in which this screening protocol will be of some, and potentially, no utility.

Identifying what proportion of the problem would likely be solved by targeting certain injuries needs to be considered. Considering how scarce training injuries occur, relative to the amount of time to accumulate 1,000 h, addressing common match-play injuries may be a more efficient endeavour for managing injury risk. Similarly, the sensitivity and specificity of screening tools may vary across the season given the nature of activities associated with specific cycles (*Bollars et al., 2014*). For instance, there is scope to reduce time-loss if lower limb (79%) or match-play (52%) injuries were addressed. Consensus on the management of common injuries is needed, even more so if the approach of frequent targeted screening during periods associated with the onset of common injuries fails. Furthermore, considering the scarcity at which some injuries occur each season, clinicians will be unlikely to statistically attribute changes in susceptibility to specific clinical entities with team interventions. The case load of a clinician across multiple seasons will not facilitate the exploration of efficace interventions for reducing and rehabilitating even the most common injuries on an elite Gaelic football team. Thus, participation in large-scale injury surveillance databases is necessary to pool sufficient quantities of quality data to monitor injury trends (*Van Dyk et al., 2017*; *Büttner, Delahunt & Roe, 2018*).

## Time for minimum standards for managing common injuries

Hamstring, knee, ankle, groin, and quadricep injuries were the most common injuries in both activities. These results mean that three out of four injuries in Gaelic football

will affect one of five lower limb sites. One in 10 match-play injuries also affected the shoulder. Thus, it is important that practitioners have a comprehensive understanding of methods to manage the assessment, diagnosis, rehabilitation, and risk management of these specific injuries. Establishing minimum reporting standards for these common injuries would reduce potential difficulties in these processes (*Delahunt et al., 2015*). These actions likely have implications for human resources operations surrounding the recruitment and development of medical and coaching staff to deliver interventions specific to the sport. Epidemiological information as presented here can also guide governing bodies in supporting medical resources at specific stages of the season associated with a greater injury rate (e.g. competitive cycles) and higher treatment costs (*Roe et al., 2016a*).

### Limitations

A major limitation of the current study is the reliance on elite teams to voluntarily participate in this injury surveillance project as it is not compulsory for teams to collect and share these data with the governing body. Thus, it is currently impossible to establish longitudinal trends in the rates of common injuries during match-play and training. The current study was also unable to investigate the relationship between player characteristics, activity workloads, and risk of sustaining a time-loss injury. This should be a priority for future research as completing screening targeted at specific injuries more frequently may address limitations in traditional approaches to injury screening while assisting in monitoring desired training responses.

## CONCLUSION AND METHODOLOGICAL CONSIDERATIONS

Each season elite Gaelic football teams will sustain 24 match-play and 15 training time-loss injuries. Regardless of activities, most injuries affect the lower limbs, are incited by non-contact injury mechanisms, are associated with an acute onset, and result in 8–28 days absence from sport. When time spent in activities is standardised to 1,000 exposure h, injuries occur 12.9-time more frequently in match-play than in training. Similarly, time-loss days per 1,000 h are 13.4-times greater in match-play than in training. The utilisation of screening tools in future studies should be targeted at seasonal cycles associated with the greatest injury risk to maximise the potential to identify high risk players.

### Funding

This work was supported by an unrestricted educational grant from the Medical, Scientific and Player Welfare Committee of the Gaelic Athletic Association. The funders had no role in study design, data collection and analysis, decision to publish, or preparation of the manuscript.

### Grant Disclosures

The following grant information was disclosed by the authors:
Medical, Scientific and Player Welfare Committee of the Gaelic Athletic Association.

## Competing Interests

John C. Murphy is a director of Medfit Proactive Healthcare.

## Author Contributions

- Mark Roe conceived and designed the experiments, performed the experiments, analyzed the data, contributed reagents/materials/analysis tools, prepared figures and/or tables, authored or reviewed drafts of the paper, approved the final draft.
- John C. Murphy conceived and designed the experiments, performed the experiments, analyzed the data, contributed reagents/materials/analysis tools, authored or reviewed drafts of the paper, approved the final draft.
- Conor Gissane conceived and designed the experiments, performed the experiments, analyzed the data, contributed reagents/materials/analysis tools, authored or reviewed drafts of the paper, approved the final draft.
- Catherine Blake conceived and designed the experiments, performed the experiments, analyzed the data, contributed reagents/materials/analysis tools, authored or reviewed drafts of the paper, approved the final draft.

## Human Ethics

The following information was supplied relating to ethical approvals (i.e., approving body and any reference numbers):

The Human Research Ethics Committee, University College Dublin granted Ethical approval (LS-E-11-91) to carry out the study within its facilities.

## Data Availability

The raw data are provided in a Supplemental File.

## Supplemental Information

Supplemental information for this article can be found online at http://dx.doi.org/10.7717/peerj.4895#supplemental-information.

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
