# Peer review of "Time to get our four priorities right: an 8-year prospective investigation of 1326 player-seasons to identify the frequency, nature, and burden of time-loss injuries in elite Gaelic football"

_PeerJ, doi:10.7717/peerj.4895_

## Round 0.1 · original submission · Major Revisions

Dear authors,

Thank you for submitting this paper to PeerJ. You will find the reviewers' reports at the foot of this email. You will see that, as well as having a number of positive things to say about the paper, they have some concerns, and have, for the present at least, advised against acceptance. However, it seems likely that, with some further work, a revised version could be acceptable, and I would like to invite you to re-submit, without commitment, a re-drafted version of your paper, paying attention to the comments and suggestions that we have made, for further review.

With respect and warm regards,
Dr Palazón-Bru (academic editor for PeerJ)

Reviewer 1 ·

Basic reporting

The management of the risk of injuries is a very important question. Understanding the extent of the injury problem is the first port of call in preventing sports injuries. The present study aimed to investigate the speed at which elite Gaelic soccer players suffer different injuries due to loss of time during the game and training activities.
The article includes sufficient introduction and background to demonstrate how the work fits into the broader field of knowledge

Experimental design

The investigation must have been conducted rigorously and to a high technical standard, in conformity with the prevailing ethical standards in the field.

Validity of the findings

The data is statistically sound and controlled.

Additional comments

After careful consideration, I have decided that the manuscript meets criteria for publication and must consider the manuscript for publication.

Reviewer 2 ·

Basic reporting

The manuscript is very well written; references used are recent and relevant. No major issues with the article structure, however, it would be important to learn from the Editor if the style used to present the references and the unstructured abstract are acceptable by the Journal standards or must be adjusted.

Experimental design

The experimental design does not raise any concerns. Nevertheless, relying on the reports from medical staffs, as the author acknowledge always leads to relevant bias. Also, it would be very interesting to have more details on the training and match loads (e.g. number of matches per week, number of training sessions, type of training sessions). Methodology is well described, however no details are given on the source of information regarding the players exposure.

Validity of the findings

In general, results and discussion are well written and the conclusions linked with the research question.
However, this is where I have some concerns regarding the manuscript:
1) although I understand that the exposure is calculated multiplying the number of hours of training or match by the number of athletes present, that is not explicit in the text;
2) also regarding the exposure, no details are given regarding the source of the information. Was data sent by the medical teams clearly indicating the number of training sessions and matches and the number of athletes present at each session/match? Or was that an estimation made by the authors?
3) Line 113 to 117: Team rates – please explain, if the overall injury rate per season is 42, and only match and training are considered, why the gap (24 match injuries and 15 training injuries = 39 injuries)? Then, Line 189: (…) reveal that teams sustain 24 match-play and 13 training injuries per season (…)
4) Line 131 to 158: Again a gap that I was not able to follow/understand:
a) Match injuries associated with the onset of 55.8% of all new injuries (…) training associated with the onset of 38.7% (…) of all new injuries
b) Match injuries associated with the onset of 59.3% of all recurrent injuries (…) training associated with the onset of 35.6% (…) of all recurrent injuries
5) Line 170 to 171: Match-play incidence (50.5 ….) was 13-times higher than training incidence….
However, Line 189: Injury incidence per 1000 exposure hours is 12.9 times greater in match-play (48.9/1000 hours)
6) A larger discussion of the study limitations should be included in the discussion and removed from the conclusions.

Additional comments

Dear author, thank you for the opportunity to read this very well written manuscript describing the characteristics of sports injuries sustained by Gaelic Football players during eight seasons. Gaelic football is not very popular outside Ireland, but the similarities to other football codes (e.g. rugby union, American football, soccer) made it very interesting for me to read your manuscript. I have expressed my concerns and I am sure that you will be able to provide the necessary clarifications to make your paper ready for publishing.

·

Basic reporting

No comment.

Experimental design

No comment

Validity of the findings

No comment

Additional comments

This paper addresses an important topic and provides some much needed information to identify injury prevention targets in Gaelic football. The inclusion of the figures was beneficial and the analysis was well completed. Below find some specific comments that I hope the authors will find useful when preparing their revisions.

---

## Round 0.2 · accepted · Accept

Dear authors,

I am happy to inform you that your paper has been accepted for publication in its current form in PeerJ.

Congratulations!

With respect and warm regards,
Dr Palazón-Bru (academic editor for PeerJ)

·

Basic reporting

No comment

Experimental design

No comment

Validity of the findings

No comment

Additional comments

Thank you to the authors for their comprehensive review. The authors have addressed all my comments and I support the publication of this article.